# Characterizing the Spatio-Temporal Variations of Urban Growth with Multifractal Spectra

**DOI:** 10.3390/e25081126

**Published:** 2023-07-27

**Authors:** Meng Fu, Yanguang Chen

**Affiliations:** Department of Geography, College of Urban and Environmental Sciences, Peking University, Beijing 100871, China; fumeng@pku.edu.cn

**Keywords:** multifractal spectrum, generalized correlation dimension, urban morphology, spatio-temporal evolution, spatio-temporal variation, dynamics

## Abstract

Urban morphology exhibits fractal characteristics, which can be described by multifractal scaling. Multifractal parameters under positive moment orders primarily capture information about central areas characterized by relatively stable growth, while those under negative moment orders mainly reflect information about marginal areas that experience more active growth. However, effectively utilizing multifractal spectra to uncover the spatio-temporal variations of urban growth remains a challenge. To addresses this issue, this paper proposes a multifractal measurement by combining theoretical principles and empirical analysis. To capture the difference between growth stability in central areas and growth activity in marginal areas, an index based on generalized correlation dimension *D_q_* is defined. This index takes the growth rate of *D_q_* at extreme negative moment order as the numerator and that at extreme positive moment order as the denominator. During the stable stage of urban growth, the index demonstrates a consistent pattern over time, while during the active stage, the index may exhibit abnormal fluctuations or even jumps. This indicates that the index can reveal spatio-temporal information about urban evolution that cannot be directly observed through multifractal spectra alone. By integrating this index with multifractal spectra, we can more comprehensively characterize the evolutionary characteristics of urban spatial structure.

## 1. Introduction

Complexity science has emerged as a prominent field within contemporary scientific development, and the application of multifractal methods has proven to be highly effective in studying complex systems. These methods have yielded significant advancements across various domains such as geology [1,2], hydrology [3], meteorology [4], physiology [5,6,7], physics [8,9], economics [10,11], music [12], and even urban geography. Urban systems are complex systems that exhibit scaling symmetry. Traditional mathematical methods fail to capture this characteristic, which necessitates the use of scaling analysis tools like fractal geometry [13,14,15]. Urban fractal research can be broadly classified into two categories: monofractal analysis and multifractal analysis. Monofractal analysis assumes that there is only one scaling process in fractals, and the growth probability of each part is equal, leading to the use of a single fractal dimension for measurement. Multifractal analysis, on the other hand, takes into account the multiple scaling processes in fractals, where the growth probabilities of different parts differ. Therefore, a set of multifractal parameters, known as the multifractal spectrum, is used to describe them. Since real fractals often have more than one scaling process, a single fractal dimension is insufficient to characterize them. A set of comparable parameters is required, which is provided by the multifractal spectrum [16].

The spatio-temporal evolution of urban morphology is a critical focus area within geography, but existing multifractal research is not extensive enough. Urban multifractal research can be classified into three spaces: real space, order space, and phase space [16]. Real space corresponds to the spatial subdivision of urban land [17], transportation [18,19,20], population [21,22,23,24], and other elements [25]; order space refers to the hierarchical subdivision of urban system [26,27]; and phase space corresponds to dynamic processes. However, compared to other fields such as economics, physiology, meteorology, and hydrology, multifractal research of phase space in urban geography is relatively lacking, which may be attributed to the lack of higher resolution time series data. On the other hand, in real space, while the spatial meaning of multifractal spectra is clear, the temporal evolution of these spectra still requires further investigation. At the spatial level, the spectra under positive moment orders correspond to central or high-density areas in a city, while those under negative moment orders correspond to marginal or low-density areas [17]. It has been observed that the spectra under positive moment orders exhibit similarities across different cities, while the spectra under negative moment orders display significant variations between cities [28,29]. This distinct growth pattern of spectra suggests that central areas represent stable zones for urban development, while marginal areas are the most dynamic regions for urban growth. At the temporal level, the multifractal parameters in the mathematical world are continuous and stable. Nevertheless, the evolution of cities in the real world, as reflected in multifractal parameters, may exhibit local fluctuations. While many studies focus on seeking macroscopic patterns of urban evolution through spectrum comparison [19,30,31,32,33,34,35] or trend fitting [36], few studies delve into characterizing spatio-temporal variations of urban evolution at a microlevel.

This paper aims to investigate the detailed variations in the spatio-temporal evolution process of urban form. To achieve this, a new measurement is introduced that captures the distinct growth characteristics of multifractal spectra under positive and negative moment orders. By utilizing case studies, the effectiveness of the analysis is verified. The following is divided into four sections. Section 2 introduces the multifractal model and case studies used in this paper, which are the Pearl River Delta (PRD) and the Yangtze River Delta (YRD), the two largest Chinese urban agglomerations. In Section 3, the conventional multifractal spectrum analysis is combined with the newly proposed measurement to depict the macroscopic laws and microvariations in the spatio-temporal evolution of urban morphology within the study areas. The findings are then compared with regional policies and spatial patterns to validate the effectiveness of the analysis. In Section 4, the achievements, novelties, and limitations of the research methods used in this paper are discussed. Finally, the main conclusions based on the research results and problem discussions are drawn.

## 2. Materials and Methods

### 2.1. Multifractal Measures

Before delving into the case analysis, let’s first introduce the multifractal measures used in this paper. Multifractals have two sets of parameters: global and local. Global parameters include generalized correlation dimension *D_q_* and mass exponent *τ_q_*, while local parameters include singularity exponent *α* and local fractal dimension *f*(*α*). Based on Renyi entropy, the calculation formula of generalized correlation dimension *D_q_* is [37,38]:(1)$$D_{q}=\left\{\begin{array}{l} \frac{1}{q-1}\underset{\varepsilon \to 0}{\lim}\frac{\ln\sum_{i=1}^{N(\varepsilon )}p_{i}{}^{q}}{\ln(\varepsilon )},q\neq 1\\ \underset{\varepsilon \to 0}{\lim}\frac{\sum_{i=1}^{N(\varepsilon )}p_{i}\ln p_{i}}{\ln(\varepsilon )},q=1 \end{array}\right.,$$ When measuring the fractal dimension using the box-counting method, *N*(*ε*) in Equation (1) represents the number of non-empty boxes under box scale *ε*, while *p_i_* represents the probability measure, that is, the ratio of measure *M_i_* in the *i*th box to the total measure *M*. The moment order *q* functions like telescopes and microscopes, and is used to highlight either the central areas (if *q* > 1 and *q*→+∞) or the marginal areas (if *q* < 1 and *q*→−∞), thereby obtaining the corresponding global dimension *D_q_* [17]. Plotting *D_q_* for different moment orders *q* on the same graph results in the spectrum of the generalized correlation dimension. The value of *D_q_* monotonically decreases with *q*, and the height difference between the upper and lower parts of the spectrum is
∆*D* = *D*_−∞_ − *D*_+∞_,(2)
representing the spatial heterogeneity and multifractality of the study area [39], with higher values indicating stronger spatial heterogeneity. Another important global parameter is the mass exponent *τ_q_*, which is related to the generalized correlation dimension through the equation
*τ_q_* = (*q* − 1)*D_q_*.(3) Since the two are somewhat equivalent, the analysis of global parameters mainly focuses on the generalized correlation dimension. On the other hand, the singularity exponent *α* of the local parameters is defined based on the probability measure *p_i_* as well [40]:(4)$$p_{i}\propto \varepsilon^{\alpha },$$
which means boxes with different probability measures have varying singularity exponents. A lower singularity exponent *α* implies a denser corresponding box, while a higher value implies less density. The singularity exponent spectrum is generated by plotting different moment orders *q* and their corresponding *α*(*q*) on the same graph, and its variation trend is generally consistent with the *D_q_* spectrum. Another important local parameter is the local dimension *f*(*α*), which represents the local fractal dimension of a subset of boxes with a given singularity exponent *α*. By plotting *α* and its corresponding local fractal dimension *f*(*α*) on the same graph, we can generate the singularity spectrum. This curve is unimodal and provides important insights into the spatial characteristics of the research area. The singularity spectrum’s maximum value point *α*_0_, and its corresponding maximum value *f*(*α*_0_), are all related to the capacity dimension *D*_0_. *D*_0_ reflects the degree of spatial filling in the research area. A larger *D*_0_ indicates a higher level of spatial filling and a more complex spatial structure. The width of the singularity spectrum, Δ*α* = *α*_−∞_ − *α*_+∞_, represents the differences between high- and low-density areas and the spatial heterogeneity of the study area. The height difference between the left and right side of the spectrum, Δ*f* = *f*(*α*_+∞_) − *f*(*α*_−∞_), reflects the fractal growth pattern of the study area. If Δ*f* > 0, high-density areas dominate and fractal growth mainly extends outward. On the contrary, if Δ*f* < 0, low-density areas dominate and fractal growth is mainly centered [41]. However, global and local parameters are essentially equivalent and can be transformed into each other through Legendre transformations which are [42]:(5)$$\begin{array}{c} \alpha (q)=\frac{d\tau (q)}{dq}\;.\\ f(\alpha )=q\alpha -\tau (q) \end{array}$$ They offer different perspectives for understanding the fractal characteristics of the study area.

There are two approaches to estimating multifractal parameters. The first is a global-to-local approach, where global parameters are estimated using Equations (1) and (3), followed by the estimation of local parameters using Legendre transformation, i.e., Equation (5). However, this method often incurs high errors due to the need for discretization in Legendre transformation of real data. The second approach is a local-to-global approach, where local parameters are calculated first, and then global parameters are inferred using Legendre transformation, i.e., Equation (5). This process does not require discretization and is therefore more accurate than the former approach. The widely used method for estimating local parameters is the normalized *μ* method [43]. The calculation steps are as follows. Firstly, define the normalized probability *μ* based on the probability measure *p_i_*:(6)$$\mu_{i}=\frac{p_{i}{}^{q}}{\sum_{i=1}^{N(\varepsilon )}p_{i}{}^{q}}.$$ Then we can then estimate the singularity exponent *α* and its local fractal dimension *f*(*α*) using
(7)$$\alpha (q)=\underset{\varepsilon \to 0}{\lim}\frac{\sum_{i=1}^{N(\varepsilon )}\mu_{i}\ln p_{i}}{\ln\varepsilon },$$
(8)$$f(q)=\underset{\varepsilon \to 0}{\lim}\frac{\sum_{i=1}^{N(\varepsilon )}\mu_{i}\ln\mu_{i}}{\ln\varepsilon }.$$

The process of parameter estimation involves different measurement and estimation methods. The probability measure *p_i_*, as shown in Equations (1), (7), and (8), is typically obtained using the box-counting method. This method is widely used for calculating multifractal parameters, with the functional box method being the most convenient form of this method [44,45]. To begin, the minimum area box of the study area is selected as the first-level box. Subsequently, each box is divided into four and then sixteen, with the side length of the box during each division corresponding to a specific scale *ε*. On the other hand, the limit condition in Equations (1), (7), and (8) is hard to meet, hence the parameter is usually approximated by the slope of the scatter plot. A variety of parameter estimation methods exist, including the ordinary least squares method (OLS), maximum likelihood method (MLM), etc. [28]. In this paper, we have chosen to use OLS and fix the intercept at 0 during regression [46].

### 2.2. Temporal Evolution of Multifractal Measures

The multifractal method provides a way to examine different density areas within a city, and the temporal evolution of multifractal parameters can reflect the growth characteristics of the city at multiple levels. Multifractal spectra under positive moment orders correspond to central areas that are stable areas of urban development. They usually grow steadily over time [19,30], reflecting the stable filling of central areas. On the other hand, multifractal spectra under negative moment orders correspond to marginal areas that are the most active areas of urban development. They usually change sharply over time, reflecting the dynamic expansion of marginal areas. The distinct evolution of multifractal parameters under positive and negative moment orders leads to significant fluctuations in Δ*D* over time, as described in Equation (2), reflecting rich spatio-temporal information. As *D*_+∞_ usually increases steadily over time, we can categorize the relative changes in *D*_+∞_ and *D*_−∞_ into three scenarios (Table 1). These scenarios can be summarized by introducing a new measurement, which is named as dimension growth rate ratio (*DGR*) in this paper:(9)$$DGR=\frac{\Delta {\left(D_{-\infty }\right)}_{t}}{\Delta {\left(D_{+\infty }\right)}_{t}},$$
whose numerator and denominator represent the changes of *D*_−∞_ and *D*_+∞_ from time *t* − 1 to time *t*, respectively. Furthermore:If *D*_+∞_ rises faster than *D*_−∞_, then Δ*D* decreases and 0 < *DGR* < 1. This type of change indicates that the growth of built-up areas is primarily due to filling in central areas, leading to a decrease in spatial heterogeneity. This direction of urban evolution is commonly observed in empirical research.If *D*_−∞_ rises faster than *D*_+∞_, then Δ*D* rises and *DGR* > 1. This type of change indicates that the growth of built-up areas is dominated by filling in marginal areas, leading to an increase in spatial heterogeneity. This marks the beginning of the expansion of marginal areas, where the fractal dimension has rapidly increased.If *D*_−∞_ decreases, then Δ*D* decreases and *DGR* < 0. This type of change indicates that the original low-density areas are rapidly filled into medium- to high-density areas, resulting in fewer new low-density areas, reduced dimensions, and spatial heterogeneity. This marks the transformation of original marginal areas into new subcentral areas.

It Is evident that *DGR* can effectively detect the spatio-temporal evolution of urban morphology. If *DGR* is greater than 1 or less than 0, it indicates that urban growth is primarily driven by expansion in marginal areas. Conversely, if *DGR* is between 0 and 1, it suggests that urban growth is predominantly focused on filling in central areas. The absolute value of *DGR* is directly proportional to the growth rate of *D*_−∞_, reflecting the intensity of the expansion process. A larger |*DGR*| signifies a faster change in *D*_−∞_ and a more significant expansion of marginal areas. To identify outliers in *DGR*, statistical methods based on the mean and standard deviation can be employed. Specifically, if a value exceeds one-standard-deviation bands, there is a 68% confidence level for determining it as an outlier. Similarly, if a value surpasses two-standard-deviation bands, a 95% confidence level is used to identify it as an outlier. The numerical detection method (0 and 1) and the standard deviation detection method complement each other and provide valuable insights into the variation details of the spatio-temporal evolution of urban morphology. An abnormal fluctuation of *DGR* in year *t* reflects the rapid expansion of marginal areas between the years *t* and *t* + 1. However, it is important to note that the former leans more towards the theoretical interpretation of *DGR*, where different values of *DGR* reflect distinct characteristics of urban morphology evolution. On the other hand, the standard deviation detection method leans more towards statistically testing *DGR*. When the index does not exhibit a clear trend or shows a weak trend, outliers can be intuitively identified using the mean and standard deviation, corresponding to significant expansion in marginal areas. In summary, this paper utilizes various multifractal measurements, including multifractal spectra and *DGR*, to comprehensively analyze the spatio-temporal evolution of urban morphology. These measurements are summarized in Table 2.

### 2.3. Study Area and Data Processing

Based on multifractal spectra and the newly defined *DGR*, the spatio-temporal evolution and variation details of urban morphology are studied below. The analysis utilizes data from two sources: annual artificial impervious area data from 1985 to 2018 [47] and urban boundaries, which represent the physical boundaries of the impervious area in pieces, every 5 years from 1990 to 2018 [48]. In order to conduct the analysis, the impervious area is cropped using the physical boundaries from 2018, then multifractal spectra, and related multifractal measurements are calculated every year based on the minimum area box of the study area. The two largest Chinese urban agglomerations, PRD and YRD, are selected as the research objects in empirical analysis; they are both comparable in terms of area size and both experienced rapid development from 1985 to 2018 (Figure 1 and Table 3). PRD encompasses 10 prefecture cities, with the central cities being Guangzhou and Shenzhen. On the other hand, YRD includes Shanghai as its central city along with six other prefecture cities. The historical background and development characteristics of the two regions make them suitable cases for studying the spatio-temporal evolution and variation details of urban morphology. The raw data are processed using ArcGIS 10.3 and ENVI 5.3, and the multifractal parameters are calculated using Python 3.8 for moment orders *q* ranging from −20 to 20.

## 3. Results

### 3.1. Multifractal Analysis

The respective multifractal spectra and related multifractal measurements of PRD and YRD are calculated from 1985 to 2018, and the results are shown in Figure 2 and Figure 3 and Appendix A. It was found that the multifractal spectra of PRD in 1985 were significantly lower as shown in Figure 2a,c, and this abnormal year has been excluded from subsequent analysis.

Based on the temporal evolution of multifractal spectra (Figure 2), we can derive the macroscopic laws in the spatio-temporal evolution of urban morphology in the study areas (Table 4). The following is a summary of these observed patterns:Generalized correlation dimension *D_q_*, as shown in Figure 2a,b:
The spectra continue to rise, indicating ongoing filling of PRD and YRD.The right side of the spectrum (*q*→+∞) converges rapidly, with an even faster convergence over time, while the left side of the spectrum (*q*→−∞) does not exhibit a clear convergence trend. This suggests that the central areas have reached a state of saturation in terms of development and filling, and there is still potential for expansion in marginal areas.Singular spectrum *f*(*α*), as shown in Figure 2c,d:
The maximum value, *f*(*α*_0_) = *D*_0_, keeps increasing, signifying an increasing degree of spatial filling.The left side of the spectrum (*q*→+∞) shows a rapid rise, while the right side (*q*→−∞) experiences a gradual decline. This implies that the central areas are rapidly being filled, and the marginal areas are expanding and transforming into new subcenters, resulting in a relatively reduced dimension.In the early stages of the study, the heights of the right side of the spectra for PRD and YRD are similar, but the left side of YRD is higher. By the end of the research period, the heights of the left side of the spectra for both regions become similar, but the right side of YRD is lower. This indicates that the central areas in YRD developed earlier compared to PRD, with a higher degree of spatial filling and more advanced development. Moreover, the expansion of marginal areas in YRD has been faster, leading to a lower dimension and more new subcenters formed during the research period. Although the central areas in PRD started their development later, they progressed rapidly and approached the filling degree observed in YRD by the end of the research period.The height difference between the left and right sides of the spectra, Δ*f*, changed from negative to positive in PRD and YRD in 2000 and 2001, respectively. This shift indicates that the fractal growth pattern of the study areas shifted from concentration to deconcentration during those corresponding years.


By analyzing the temporal evolution of *DGR* (Figure 3), we can uncover the microscopic variations in the spatio-temporal development of urban morphology in the study areas (Table 4). The generalized correlation dimension *D_q_* exhibits a steady increase under positive moment orders, eventually reaching a stabilized state in the later stages. Furthermore, there is no significant difference observed between PRD and YRD. Under negative moment orders, *D_q_* generally increases, but with notable fluctuations. Particularly significant fluctuations are observed in 2009 in PRD and in 1997 in YRD (Figure 3a,b). The distinct evolution of *D_q_* under positive and negative moment orders leads to significant fluctuations in Δ*D* (Figure 3c,d). Overall, Δ*D* decreases, indicating a decreasing trend in spatial heterogeneity within the study areas. However, Δ*D* for PRD fluctuated significantly in 2009 and 2016, while those for YRD experienced significant fluctuations in 1997, 2015, and 2016, reflecting fluctuations in spatial heterogeneity during the respective years. The aforementioned spatio-temporal variations in urban morphology evolution can be further highlighted by examining the temporal evolution of *DGR*, providing a more intuitive explanation (Figure 3e,f). In most cases, *DGR* remains stable between 0 and 1, indicating a consistent filling of central areas. However, there are certain years where *DGR* shows abnormal fluctuations, surpassing 0 or 1, and even exhibiting sudden jumps. These abnormal fluctuations indicate active expansion in marginal areas. The abnormal fluctuations in *DGR* can occur at a single point in time or over a period of time. The former marks an important node in the expansion of marginal areas, such as 1994 and 1997 in YRD. The latter marks a period of rapid expansion of marginal areas, namely, the expansion of marginal areas (*DGR* > 1)→the original marginal areas becoming new subcentral areas (*DGR* < 0)→the expansion of new marginal areas (*DGR* > 1)… This type of expansion is particularly evident during 2005 to 2010 and 2013 to 2017 in PRD, as well as 2013 to 2016 in YRD. To detect outliers in *DGR*, statistical criteria can be established using standard-deviation bands. Applying one standard deviation, we can determine “abnormal” years in PRD as 2009, 2010, 2013, 2014, 2016, and 2017, and in YRD as 1994, 1997, 2013, 2015, and 2016, at a 68% confidence level. Moreover, based on two standard deviations, “abnormal” years in PRD are 2009 and 2016, and in YRD are 1997, 2015, and 2016, at a 95% confidence level. This standard deviation detection method allows for the identification of outliers in *DGR*.

The abnormal fluctuations in the *DGR* curve over time indicate active expansion of marginal areas in the study areas. The standard deviation detection reveals that YRD has started rapid growth in its marginal areas since 1994, whereas PRD encountered a similar situation only in 2009. This suggests that the urban development in YRD generally preceded that of PRD, which is consistent with the results from multifractal spectra. Despite PRD being known as a window for China’s reform and opening-up policies, YRD holds a longstanding position as an established economic region within China.

### 3.2. Policy Analysis

The active expansion of marginal areas may be attributed to regional or national policies. In 2008, the Outline of the Plan for the Reform and Development of the Pearl River Delta (2008–2020) was approved, aiming to promote regional economic integration and deepen cooperation in the Pan-PRD area. The provincial government aims to achieve “A good start in one year, a great development in four years, and a great leap in nine years”, so 2009, 2013, and 2017 are key years for the expansion of PRD. In the case of YRD, the establishment of the Seminar on the Work of the Yangtze River Delta Political Consultative Conference in 1994 and the Inter-City Conference on Yangtze River Delta Economic Coordination in 1997 laid the foundation for political and economic cooperation in the region. And the establishment of the China (Shanghai) Pilot Free Trade Zone in 2013 provided further impetus for the integration of YRD. The integrated development of these regions also relies on the support of national policies. In 2016, the 13th Five-Year Plan, adopted during the Fourth Session of the 12th National People’s Congress, aimed to optimize and enhance the urban agglomerations in the eastern region, including Beijing–Tianjin–Hebei, YRD, and PRD. It also emphasized the coordinated development of the upper, middle, and lower reaches of the Yangtze River and regional cooperation in the Pan-PRD, supporting and facilitating further expansions of YRD and PRD. While the implementation of the 13th Five-Year Plan may not have had an immediate impact on urban morphology, the policy of further expansion in the Pan-YRD and Pan-PRD regions indirectly indicates that the expansion of the YRD and PRD regions has already commenced and has reached a certain extent. Consequently, the *DGR* of YRD and PRD both fluctuated significantly in 2016, corresponding to a significant expansion of marginal areas from 2016 to 2017. As a result of the expansion in marginal areas, multiple cities have seen the interconnection of their built-up areas. In PRD, the marginal areas of southern Guangzhou and western Dongguan expanded significantly from 2005 to 2010, leading to an interconnection of the built-up areas of Guangzhou, Foshan, Dongguan, and Shenzhen, forming an integrated spatial pattern (Figure 4a,b). From 2010 to 2018, Zhongshan and Zhuhai also expanded and connected with the built-up areas of these cities, further consolidating the integrated spatial pattern (Figure 4b,c). In YRD, the marginal areas of eastern Changzhou, northern Wuxi, Suzhou, and Shanghai significantly expanded from 2010 to 2015, leading to an interconnection of built-up areas of these cities for the first time. As a result, the integrated spatial pattern was initially formed (Figure 5a,b). From 2015 to 2018, the marginal areas of Shanghai continued to expand, further strengthening the integration pattern (Figure 5b,c). In summary, *DGR* can effectively reflect the spatio-temporal variation of urban evolution.

## 4. Discussion

The multifractal method is a valuable approach for analyzing the spatio-temporal evolution of urban morphology. However, there is still a need to explore the specific details of urban evolution. Therefore, this paper proposes a multifractal measurement called the dimension growth rate ratio (*DGR*) to characterize the spatio-temporal variation of urban evolution, and it is validated using PRD and YRD as examples. Based on the above calculations and analysis, the research results can be summarized as follows. Firstly, the generalized correlation dimension (*D_q_*) shows a steady increase under positive moment orders and significant fluctuations under negative moment orders. The differences between the two reflect rich spatio-temporal information. *D_q_* under positive moment orders are similar among different cities, demonstrating a consistent increase over time and stabilization in later stages, indicating the stable filling of central areas. On the other hand, *D_q_* under negative moment orders generally increase over time, but with notable fluctuations, reflecting the active expansion of marginal areas. Based on this, the spatio-temporal evolution pattern of urban morphology can be classified into three scenarios: (1) *D*_+∞_ rises faster than *D*_−∞_, indicating urban growth dominated by filling in central areas, corresponding to the common direction of urban evolution. (2) *D*_−∞_ rises faster than *D*_+∞_, indicating urban growth dominated by filling in marginal areas, corresponding to the beginning of marginal area expansion. (3) *D*_+∞_ increases while *D*_−∞_ decreases, suggesting rapid filling of original low-density areas, resulting in fewer new low-density areas and a smaller fractal dimension. This corresponds to the transformation of the original marginal areas into new subcentral areas. Secondly, the spatio-temporal variation of urban evolution can be quantitatively detected using *DGR*. *DGR* represents the ratio of the growth rate between *D*_−∞_ and *D*_+∞_ and can comprehensively reflect the evolution of both central and marginal areas. Normally, 0 < *DGR* < 1 indicates stable growth in central areas. Abnormal fluctuations in *DGR* reflect the expansion of marginal areas. *DGR* > 1 corresponds to the beginning of expansion in marginal areas, while *DGR* < 0 corresponds to the transformation of original marginal areas into new subcentral areas. In some cases, *DGR* may even exhibit abrupt jumps. Larger |*DGR*| indicates more significant expansion and can be tested using standard-deviation bands. Thirdly, abnormal fluctuations in *DGR* can occur at a single time point or over a certain period of time. A single time point with abnormal fluctuation signifies a crucial node for marginal area expansion, while fluctuations over a period of time mark periods of rapid expansion.

By combining the conventional multifractal spectra with the proposed *DGR* measurement, important conclusions can be drawn regarding the spatio-temporal evolution patterns in PRD and YRD. Overall, from 1985 to 2018, both PRD and YRD experienced an increase in the degree of spatial filling and complexity. The central areas reached a saturation point, while there is still potential for expansion in marginal areas. Compared to PRD, the urban growth in YRD occurred earlier and was more developed. YRD exhibited a higher level of filling in its central areas and witnessed faster expansion in the marginal areas. Nevertheless, PRD demonstrated rapid development, and by 2018, the degree of filling in its central areas approached that of YRD. Moreover, in PRD, the pattern of fractal growth shifted from concentration to deconcentration in 2000. Marginal areas expanded rapidly from 2005 to 2010 and from 2013 to 2017, with notable growth observed in 2009 and 2016, leading to the formation and strengthening of regional integration patterns. In YRD, significant expansion nodes occurred in 1994 and 1997, with policies laying the political and economic foundations for regional integration. The pattern of fractal growth shifted from concentration to deconcentration in 2001. The period from 2013 to 2016 witnessed another rapid expansion in marginal areas, indicating the formation and strengthening of regional integration patterns, with notable growth observed in 2015 and 2016. The key nodes and periods in the evolution of urban form may be attributed to regional or national policies and are reflected in spatial patterns.

The spatio-temporal evolution of urban systems is a crucial aspect of urban planning, and multifractal methodology offers an effective tool for studying this evolution. However, due to limited time series data, most studies have focused on comparing changes in multifractal spectra to understand the general laws of urban development [19,30,31,32,33,34,35], or fitted the time series of *D_q_* using logistic functions to reveal the spatial replacement dynamics of urban development. Based on logistic functions, it is possible to determine the type of urban evolution, predict the time of maximum speed, and divide urban development stages macroscopically [36]. However, these studies have primarily remained at a macrolevel analysis of urban evolution, with few exploring the microvariations within the evolution process. Compared with previous research, this article introduces an innovative approach by employing a newly defined measurement called *DGR* to detect the detailed spatio-temporal variations in urban evolution. The different values of *DGR* correspond to the expansion of central or marginal areas, and abnormal fluctuations in these values indicate active expansion in marginal areas. The calculation of *DGR* is clear, concise, and useful for high temporal resolution data. It could serve as a supplement to previous macro-analysis and provide more insights into the spatio-temporal evolution patterns of urban morphology.

This paper has several shortcomings that need to be addressed. Firstly, the algorithm is limited to OLS with a fixed intercept of 0, which might be not comprehensive enough. The parameter estimation of mathematical models depends on algorithms, while the selection of algorithms is subjective. The commonly used algorithms for estimating regression parameters include OLS and MLM. An empirical study showed that when the observed data follow a power law, the results of the two algorithms are consistent; when the observation data do not obey a power law, OLS gives an approximate value, while MLM gives an outlier. Therefore, MLM can be used to detect power laws [28]. Furthermore, OLS for parameter estimation also has two possible scenarios: fixed intercept and unfixed intercept. Previous research indicates that the results from these two methods are consistent when the fractal structure of cities is well-developed. However, if the fractal structure of cities is not sufficiently developed, multifractal measurement depends on the selection of methods [46]. Therefore, integrating multiple algorithms and methods, including both OLS and MLM, or both fixed and unfixed intercepts, can aid in understanding the evolution of urban fractal structure. Unfortunately, due to limited space, this paper only chose to utilize OLS with a fixed intercept for parameter estimation. Future comparative analysis measuring *DGR* under different algorithms and methods may improve our understanding of urban evolution. Secondly, this research only covers a limited number of cities; the universality of the measurement *DGR* needs further verification, and the mechanism behind the spatio-temporal variation in urban evolution still needs further exploration.

## 5. Conclusions

The classic theme in geography focuses on regional differentiation, and geographers are interested in understanding both spatial differences and their underlying similarities. Multifractal methodology offers a way to unify regional differentiation and spatial similarities within a single descriptive framework. When the moment order is greater than 0, the multifractal spectrum provides more information about central areas with high density and growth probability, as well as intra-area stability and inter-area similarity. On the other hand, when the moment order is less than 0, the multifractal spectrum captures more information about marginal areas with low density and growth probability, along with intra-area instability and inter-area differences. Based on these ideas, this article introduces an index using extreme values of the generalized correlation dimension, named as *DGR*, to describe the spatio-temporal evolution characteristics of cities. This index combines two extremes and can effectively capture the detailed features of urban spatio-temporal evolution. Through calculations, analysis, and a discussion of the problems, the following main conclusions can be drawn. Firstly, the curve of the growth rate of the generalized correlation dimension under extreme positive and negative moment orders (*DGR*) over time reflects the variation characteristics of urban growth. Urban growth is more active in marginal areas and can be characterized by multifractal parameters under negative moment orders. Conversely, growth in urban central areas is relatively stable and can be characterized by multifractal parameters under positive moment orders. The ratio of the growth rate of the generalized correlation dimension between these two extreme cases reflects the growth rate of marginal areas relative to central areas. A *DGR* greater than 1 or less than 0 corresponds to active expansion of marginal areas. Abnormal fluctuations in the *DGR* curve over time typically indicates a period of active growth in urban marginal areas. Secondly, the *DGR* curve serves as a complementary tool of multifractal analysis for urban morphology. In other words, combining the *DGR* curve with multifractal spectra allows for a more comprehensive understanding of urban growth. The generalized correlation dimension and related multifractal measurements can effectively reflect the spatio-temporal evolution patterns of urban morphology. The stable filling in central areas represents a normal situation, while expansion in marginal areas indicates an “abnormal” situation, which corresponds to sudden increases or decreases in parameters. The *DGR* curve captures these abnormal growth patterns and changes. By using various multifractal parameter curves, urban growth characteristics can be comprehensively analyzed from multiple perspectives and levels.

## Figures and Tables

**Figure 1 entropy-25-01126-f001:**
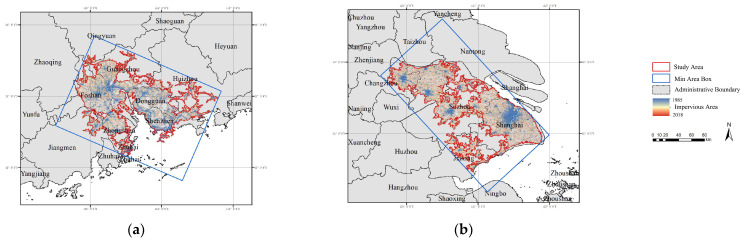
Study area of the research objects: (**a**) the Pearl River Delta (PRD); and (**b**) the Yangtze River Delta (YRD). The two are the largest urban agglomerations in China.

**Figure 2 entropy-25-01126-f002:**
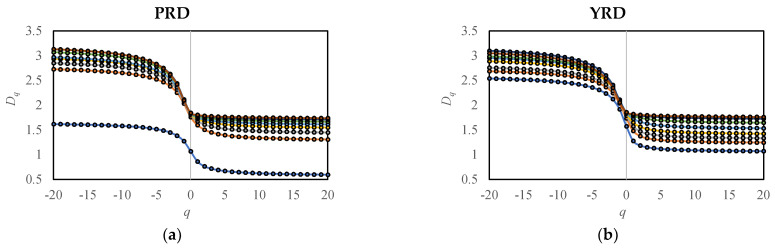
Multifractal spectra every five years from 1985 to 2018 in PRD and YRD: (**a**) generalized correlation dimension spectrum *D_q_* in PRD; (**b**) generalized correlation dimension spectrum *D_q_* in YRD; (**c**) singularity spectrum *f*(*α*) in PRD; and (**d**) singularity spectrum *f*(*α*) in YRD.

**Figure 3 entropy-25-01126-f003:**
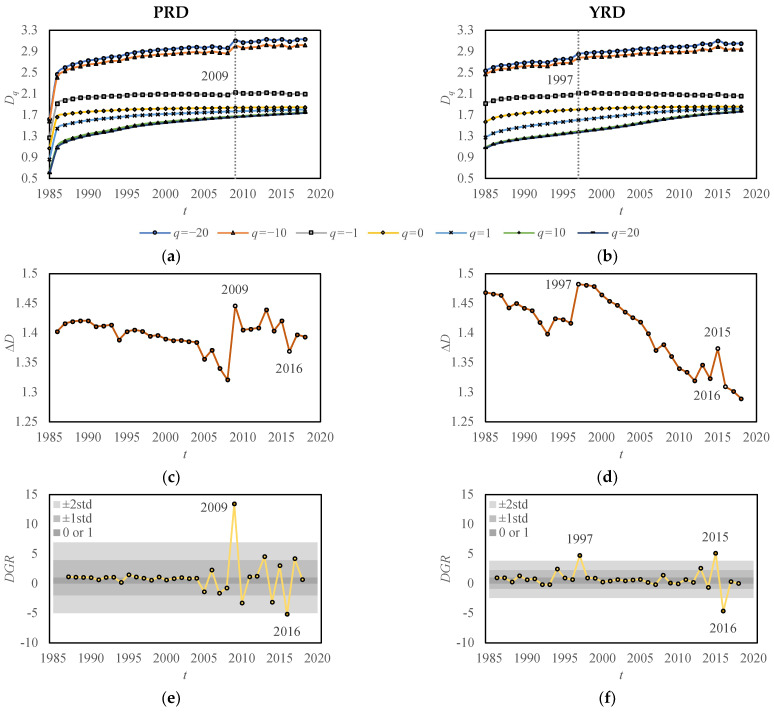
Temporal evolution of related multifractal measurements from 1985 to 2018 in PRD and YRD: (**a**) generalized correlation dimension *D_q_* in PRD; (**b**) generalized correlation dimension *D_q_* in YRD; (**c**) height difference ∆*D* in PRD; (**d**) height difference ∆*D* in YRD; (**e**) newly defined measurement *DGR* in PRD; and (**f**) newly defined measurement *DGR* in YRD. In subfigures (**e**,**f**), the light-gray and medium-gray bands represent the two-standard-deviation and one-standard-deviation bands, respectively. If a data point falls outside the bands, there is a 95% (light-gray) or 68% (medium-gray) confidence level for considering it as an outlier. The dark-gray boundaries are set at 0 and 1. When a data point goes beyond these boundaries, it indicates that urban growth is mainly driven by expansion in marginal areas. On the contrary, it suggests that urban growth is dominated by filling in central areas.

**Figure 4 entropy-25-01126-f004:**
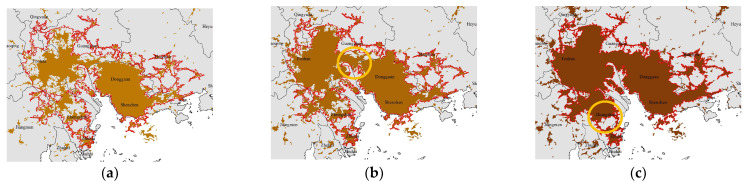
Temporal evolution of physical boundaries in PRD from 2005 to 2018: (**a**) physical boundaries in 2005; (**b**) physical boundaries in 2010; and (**c**) physical boundaries in 2018. Southern Guangzhou and western Dongguan highlighted by the circle in subfigure (**b**) experienced rapid expansion from 2005 to 2010. Zhongshan and Zhuhai highlighted by the circle in subfigure (**c**) experienced rapid expansion from 2010 to 2015.

**Figure 5 entropy-25-01126-f005:**
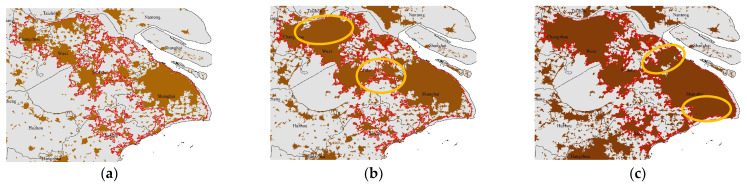
Temporal evolution of physical boundaries in YRD from 2010 to 2018: (**a**) physical boundaries in 2010; (**b**) physical boundaries in 2015; and (**c**) physical boundaries in 2018. Eastern Changzhou, northern Wuxi, Suzhou, and Shanghai highlighted by the circles in subfigure (**b**) experienced rapid expansion from 2010 to 2015. The marginal areas of Shanghai highlighted by the circles in subfigure (**c**) experienced rapid expansion from 2010 to 2015.

**Table 1 entropy-25-01126-t001:** The classification of the temporal evolution of the generalized correlation dimension and related multifractal measurements.

Change of *D_q_*	Change of Related Measurements	Spatio-Temporal Information
∆(*D*_+∞_)*_t_* > ∆(*D*_−∞_)*_t_* > 0	Δ*D* decreases 0 < *DGR* < 1	Dominated by filling in central areas, spatial heterogeneity decreased: common direction of urban evolution.
∆(*D*_−∞_)*_t_* > ∆(*D*_+∞_)*_t_* > 0	Δ*D* rises *DGR* > 1	Dominated by filling in marginal areas, spatial heterogeneity increased: the beginning of expansion of marginal areas.
∆(*D*_+∞_)*_t_* > 0 > ∆(*D*_−∞_)*_t_*	Δ*D* decreases *DGR* < 0	The original low-density areas filled rapidly, new low-density areas reduced and fractal dimension decreased, spatial heterogeneity decreased: the transformation of original marginal areas into new subcentral areas.

**Table 2 entropy-25-01126-t002:** Main multifractal measurements used in this paper.

Measurement	Meaning in Spectrum	Spatio-Temporal Information
Δ*D* = *D*_−∞_ − *D* _+∞_ Δ*α* = *α*_−∞_ − *α*_+∞_	Height difference between the upper and lower parts of the *D_q_*, *α*(*q*) spectrum.	Differences between high- and low-density areas: the higher, the stronger the spatial heterogeneity.
*f*(*α*_0_) = *D*_0_	Maximum value of *f*(*α*).	Degree of spatial filling: the larger, the higher the level of spatial filling and the more complex the spatial structure.
Δ*f* = *f*(*α*_+∞_) − *f*(*α*_−∞_)	Height difference between left and right side of *f*(*α*): >0, higher on the left; <0, higher on the right.	Fractal growth pattern: >0, high-density areas dominate, and fractal growth mainly extends outward; <0, low-density areas dominate, and fractal growth is mainly centered.
*DGR* = ∆(*D*_−∞_)*_t_*/∆(*D*_+∞_)	Ratio of growth rate of *D_q_* at extreme positive and negative moment orders.	Variation details in the spatio-temporal evolution of urban morphology: if the curve exhibits steadiness over time, central areas grow steadily; if the curve exhibits abnormal fluctuations, marginal areas expand actively.

**Table 3 entropy-25-01126-t003:** Research objects: the Pearl River Delta (PRD) and the Yangtze River Delta (YRD).

Research Objects	The Pearl River Delta (PRD)	The Yangtze River Delta (YRD)
Cities included	Ten prefecture cities: Guangzhou, Shenzhen, Foshan, Dongguan, Zhongshan, Zhuhai, Huizhou, Qingyuan, Jiangmen, and Zhaoqing.	One municipality: Shanghai; six prefecture cities: Suzhou, Wuxi, Jiaxing, Changzhou, Nantong, and Zhenjiang.
Area/km^2^	10,711.8	10,480.2
Historical background and development characteristics	The central cities, Guangzhou in PRD and Shanghai in YRD, have long histories. But the entire region only started to develop rapidly since the reform and opening up. The period between 1985 and 2018 was notable for its significant and speedy progress.	

**Table 4 entropy-25-01126-t004:** Multifractal analysis results of the spatio-temporal evolution of urban morphology in PRD and YRD from 1985 to 2018.

	Temporal Evolution of Multifractal Spectra and Related Measurements	Spatio-Temporal Information in the Evolution of Urban Morphology
Macroscopic laws	*D_q_* rises, *f*(*α*_0_) = *D*_0_ increases.	Increasing degree of spatial filling.
	When *q*→+∞, *D_q_* converges rapidly, with an even faster convergence over time, while when *q*→−∞, *D_q_* does not exhibit a clear convergence trend.	Central areas have reached saturation in terms of development and filling, and there is still potential for expansion in marginal areas.
	When *q*→+∞, *f*(*α*) shows a rapid rise, while when *q*→−∞, *f*(*α*) experiences a gradual decline.	Central areas were rapidly being filled, and marginal areas were expanding and transforming into new subcenters, resulting in a relatively reduced dimension.
	In the early stages of the study, *f*(*α*_−∞_) for PRD and YRD were similar, but *f*(*α*_+∞_) for YRD was larger. By the end of the research period, *f*(*α*_+∞_) for both regions became similar, but *f*(*α*_−∞_) for YRD was smaller.	The central areas in YRD developed earlier compared to PRD, with a higher degree of spatial filling and more advanced development. The expansion of marginal areas in YRD has been faster, leading to more new subcenters formed during the research period. Although the central areas in PRD started their development later, they progressed rapidly and approached the filling degree observed in YRD by the end of the research period.
	Δ*D* and Δ*α* decreased overall.	A decreasing trend in spatial heterogeneity.
	Δ*f* changed from negative to positive in 2000 in PRD, and in 2001 in YRD.	Fractal growth pattern shifted from concentration to deconcentration in 2000 in PRD, and in 2001 in YRD.
Microscopic variations	*D_q_*_>0_ increased steadily over time, stabilized in the later stages, and exhibited similarities between PRD and YRD. *D_q_*_<0_ generally increased but sometimes fluctuated.	Central areas were steadily filled and tended to saturation, marginal areas expanded actively.
	*DGR* surpassing 0 or 1: 2005–2010, 2013–2017 in PRD; 1994, 1997, 2013–2016 in YRD.	Marginal areas expanded actively in the corresponding years, leading to the interconnection of built-up areas among multiple cities.
	When *DGR* exceeds one-standard-deviation bands, the value is an outlier with a 68% confidence level: 2009, 2010, 2013, 2014, 2016, 2017 in PRD; 1994, 1997, 2013, 2015, 2016 in YRD.	The marginal areas witnessed substantial expansion during those specific years, which might be attributed to regional or national policies. YRD started rapid growth in its marginal areas beginning in 1994, whereas PRD encountered a similar situation only in 2009. The urban development in YRD generally preceded that of PRD. Despite PRD being known as a window for China’s reform and opening-up policies, YRD holds a longstanding position as an established economic region within China.
	When *DGR* exceeds two-standard-deviation bands, the value is an outlier with a 95% confidence level: 2009 and 2016 in PRD; 1997, 2015, 2016 in YRD.	

## Data Availability

The data presented in this study are available in Appendix A.

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
