# Peer review of "Characterizing the Spatio-Temporal Variations of Urban Growth with Multifractal Spectra"

_entropy, 2023, doi:10.3390/e25081126_

Round 1

Reviewer 1 Report

In this paper, the authors addressed the problem of quantifying the dynamics of urban growth. The authors propose a methodology based on multifractal formalism to determine the structure of the urban spatial structure evolution. 

In my opinion, the paper is clear and well-written. The proposed method is well explained and justified. Moreover, the subject could be potentially interesting to the readers.

I have only some minor remarks:

- Can you justify the following statement and cite appropriate references in the main text? It is not clear to me.

"Multifractal parameters under positive moment orders primarily capture information about central areas with relatively stable growth, while those under negative moment orders mainly re-10 flect information about marginal areas with more active growth"

- To underline the interdisciplinary of multifractal formalism, more papers with MF analysis referring to diverse areas of science should be cited, e.g.:

(neuroscience)  Ochab et al. Scientific Reports (2022) 12:17866,  https://doi.org/10.1038/s41598-022-21375-1

(physiology)  Ivanov et al. Nature, vol. 399, no. 6735, pp. 461–465, 1999. 

doi: 10.1038/20924

(physics) Subramaniam et al. Physical Review B, vol. 78, article 245105, 2008. https://doi.org/10.1103/PhysRevB.78.245105

(music) Jafari et al., Journal of Statistical Mechanics: Theory and Experiment, vol. 2007, no. 4, article P04012, 2007. 

DOI 10.1088/1742-5468/2007/04/P04012

Suggestion to authors.

To quantify the dynamics of the asymmetry of the spectrum, you can use the asymmetry coefficient: 

Drozdz et al. "Detecting and interpreting distortions in hierarchical organization of complex time series", PHYSICAL REVIEW E 91, 030902(R) (2015), DOI: 10.1103/PhysRevE.91.030902

Reviewer 2 Report

1. Introduction

lines 65-69 and 66-71 the same content! Should be improved.

3. Results

I suggest dividing the content into 2 subsections - the results arising from the analysis of Multifractal spectrums and related multifractal measurements (line 227-308) and the results arising from the references to PRD and YRD development policy reforms (line 309-344).

345-348 - these are conclusions and should be in the Conclusions section.

"In 2016, the 13th Five-Year Plan, adopted during the Fourth Session of the 12th National People's Congress, aimed to optimize and enhance the urban agglomerations in the eastern  region, including the Beijing-Tianjin-Hebei, YRD, and PRD. ..... Therefore, the DGR of YRD and PRD both fluctuated significantly in 2016, corresponding to a significant expansion of marginal areas." - Did the really visible spatial changes occur as early as the year of adoption of the five-year plan?Or were they the result of other conditions or earlier decisions?

Figure 4. Temporal evolution of urban boundaries in PRD from 2005 to 2018. - Is the term "urban boundary" appropriate? Have there been changes in the course of city boundaries resulting from changes in the administrative division of units, or are these changes in urbanized areas within existing city boundaries? This question should be clarified and referred to in the text of Section 3.

4. Discussion

I propose to separate from the discussion the section on research limitations and future research (422-441).

Author Response

Please the attached file.

Round 2

Reviewer 1 Report

I support the publication of the manuscript in its present form.

Reviewer 2 Report

The text has been improved in accordance with the reviewer's comments. Most of the proposed modifications and additions have been included. The text may be published in the current version.